# A qualitative analysis of women's postnatal experiences of breastfeeding supports during the perinatal period in Ireland

**Niamh Lawlor**[1], **Lucia Prihodova**[2], **Deborah Byrne**[2], **Megan Etherton**[2], **Felicienne Rahill**[2], **Catie Wilson**[2], **Elizabeth J. O'Sullivan**[1]*

**1** School of Biological, Health and Sports Sciences, Technological University Dublin, Dublin, Ireland,
**2** Bainne Beatha, Parent-led Breastfeeding Advocacy Group, Dublin, Ireland

* liz.osullivan@tudublin.ie

**Data Availability Statement:** Data cannot be shared publicly because some participants may be identifiable depending on the nature of the qualitative data they have shared. In addition, some

## Abstract

Ireland has among the lowest rates of breastfeeding worldwide. Despite policies to support breastfeeding, breastfeeding initiation and exclusivity remain low in Ireland. Greater knowledge about support received in the maternity unit may—in part—shed light on why this is so. Our aim was to analyse women's experiences of the breastfeeding supports available in the early postnatal period in Ireland. We conducted an analysis of an open-ended question on a cross-sectional survey about breastfeeding support conducted in the Republic of Ireland in 2022. Participants were asked to provide comments about the breastfeeding support they received in the maternity unit or during your home birth. Data were analysed using Braun and Clarke's six-step Thematic Analysis Framework. There were 5,412 unique responses to the survey and 2,264 responses to the question of interest. Two themes were generated from the data: (i) 'Breastfeeding support in theory but not in practice.' Although breastfeeding was promoted by healthcare professionals antenatally, breastfeeding challenges were rarely mentioned. Participants then felt unsupported in overcoming challenges postnatally. (ii) 'Support was either inaccessible due to lack of staff/time, inadequate; i.e., unhelpful or non-specific, and/or physically inappropriate.' Most participants described receiving supports that were less than optimal in aiding them to establish breastfeeding. While many described difficulties in accessing supports, others found support to be 'non-specific,' 'rushed' and sometimes 'rough.' A lack of knowledge, time and support from healthcare professionals was frequently described, which was often recognised as a failing of the healthcare system. Women require practical, informative, and specific breastfeeding support. Barriers such as lack of time and trained staff in the maternity unit need to be addressed.

## Introduction

The World Health Organisation (WHO) recommends exclusive breastfeeding for the first 6 months of life [1]. Breastfed infants have a decreased risk of gastrointestinal and respiratory diseases and enhanced cognitive development when compared with those fed infant formula

respondents provided names and working locations of care providers in their qualitative comments. Data are available from the TU Dublin Research Ethics and Integrity Committee (contact via Elizabeth J O'Sullivan) for researchers who meet the criteria for access to confidential data.

**Funding:** The author(s) received no specific funding for this work.

**Competing interests:** The authors have declared that no competing interests exist.

[2]. Women who breastfeed are less likely to develop reproductive cancers, such as breast and ovarian cancer, diabetes mellitus and obesity [3]. Despite this, initiation rates of breastfeeding in Ireland are among the lowest worldwide [4]. In 2020, only 62.3% of women in Ireland initiated breastfeeding and on hospital discharge in 2020, at approx. day 2 of life, 36.7% of infants were exclusively breastfeeding (i.e., had not introduced infant formula) [5]. Initiation of breastfeeding is considerably lower in Ireland than other countries; over 90% of mothers initiate breastfeeding in Sweden [6] and Norway [7], and over 83% initiate breastfeeding in the United States of America [8].

Internationally, the Baby-Friendly Hospital Initiative (BFHI) and the Code of Marketing of Breast-milk Substitutes are standards in place that can protect, promote, and support breastfeeding. In Ireland, pregnant women who are ordinarily resident can access maternity care for free, under the Maternity and Infant Care Scheme. First-time pregnant women are offered a series of antenatal classes, which typically includes a breastfeeding class, though attendance at this class is not mandatory [9]. Pregnant women receive care from a general practitioner of their choice and a hospital obstetrician, from midwives and obstetricians in the maternity unit, and from their general practitioner and public health nurse postpartum [10]. On average, women stay two to three days in the maternity unit postpartum [11]. Midwives act as primary care providers for women during pregnancy, birth, labour and the post-natal period [12] and they are trained to provide breastfeeding assistance and support, per the competencies outlined by the Nursing and Midwifery Board of Ireland [13]. However, staff shortages are consistently reported in maternity units [14–16], which likely limits the time available for midwives to provide breastfeeding-related care and support.

In the maternity unit, International Board-Certified Lactation Consultants (IBCLCs) are in place to provide postnatal support when issues arise; however, not all maternity units in Ireland currently employ a full-time IBCLC. Additionally, their work hours are often limited to typical business days and hours, meaning their services are not universally accessible. This results in many women seeking private consultation from an IBCLC [17]. Upon discharge from the maternity unit, mothers typically receive a visit from their Public Health Nurse within 72 hours when they may help with feeding and information about local breastfeeding support groups [18]. The public health nurse conducts routine checks with children for the first 5 years of life.

Though standard care includes midwives and lactation consultants providing support in the maternity unit, and public health nurses providing support postpartum, this standard care was severely impacted during the COVID-19 pandemic. Visits from partners, and their attendance at antenatal appointments was restricted, and partners were only allowed be in the delivery room once the mother was in established labour [19]. Some antenatal classes and services were suspended or moved online (including breastfeeding classes) [20]. Results of an independently conducted survey suggest that mothers found these changes/restrictions stressful and isolating, especially the exclusion of partners, which impacted access to practical support with breastfeeding [21]. Stress and feelings of isolation associated with changes in the standard of care due to COVID-19 were also reported by the authors of an in-depth qualitative interview study among 19 women who gave birth during the pandemic in Ireland [19]. Similarly, Panda and colleagues reported that women struggled with limited access to in-person breastfeeding support [19]; however, others found breastfeeding easier because they had more privacy with fewer people on the wards.

In terms of international standards, the BFHI set up by the WHO and the United Nations Children's Fund (UNICEF) incorporates 'The Ten Steps to Successful Breastfeeding' [22]—a multifaceted approach to improve the quality of breastfeeding supports within the maternity unit. Increased rates of breastfeeding have been observed in facilities that implement the Ten

Steps [23, 24]. Unfortunately, in 2017 funding was withdrawn from the Baby Friendly Health Initiative in Ireland [25], which incorporated the Ten Steps as a performance indicator, meaning its implementation throughout Irish maternity facilities was brought to an end. However, the Irish Health Service Executive released a report in 2022, detailing plans to re-introduce the BFHI as part of the National Standards for Infant Feeding in Maternity Services, which should see increased breastfeeding supports available for families in Irish maternity units [26].

Another international standard that can be followed in maternity services is the WHO's International Code of Marketing of Breast-milk Substitutes, which is also included in the BFHI [24]. The Code states that there should be no advertising, promotion or free samples of breast-milk substitutes offered to mothers or their families [27]. An Irish research report released in 2020 exploring the factors influencing infant feeding decisions noted that free infant formula was being offered in the maternity unit as a solution to breastfeeding issues and that although approximately half of the participants in this research intended to exclusively breastfeed, only 38% were doing so at 72 hours post-birth [28]. In October 2021, the Health Service Executive (HSE) approved a policy on the Marketing of Breast Milk Substitutes [27] and have produced a guide for staff to facilitate them to work within the Code [29].

In countries where breastfeeding is not the social norm, such as in Ireland, support can aid women's confidence in breastfeeding [30]. Women's experiences of initiating feeding shortly after birth in Ireland have been described by Murphy and colleagues [17], using data from the National Maternity Experience Survey (NMES) [31]. Murphy and colleagues reported that women described feeling shamed and disrespected by healthcare professionals (HCPs) for their feeding choices, but that confidence increased when women received guidance from HCPs, in particular midwives, making their breastfeeding journey easier [17]. A Cochrane review by McFadden and colleagues [30] to describe different forms of breastfeeding support found that support interventions that are delivered face-to-face have the greatest impact on rates of exclusive breastfeeding [30], emphasising how women value personal interactions with HCPs. Breastfeeding support groups have been deemed valuable also; results of a qualitative interview study among 15 breastfeeding mothers in Ireland highlight that mothers value support groups for the social interaction they provide, and for normalising breastfeeding in a country with a high use of infant formula [32]. With short stays in the maternity unit common in Ireland, the need for supports to be timely and personal, and extending across the continuum from prenatal to postnatal is evident; this could play an essential role in supporting women to successfully breastfeed in Ireland.

At present, the supports and policies in place throughout the Irish healthcare system are less than optimal for assisting women in the initiation and continuation of breastfeeding. The present study aims to describe women's personal experiences of supports for breastfeeding within the Irish healthcare system during their time in the maternity unit, or during their home birth in Ireland to provide evidence for how support can be improved.

## Methods

### Study design

The present study is a qualitative analysis of an open-ended question on a national, cross-sectional survey. The study was conducted by researchers in Technological University Dublin (TU Dublin) in collaboration with Bainne Beatha, a breastfeeding advocacy group. Ethical approval for the project was obtained from the TU Dublin Research Ethics Committee (REIC-21-83). To maintain anonymity of respondents, no identifiable information was collected, e.g., names, email addresses, IP addresses. The last author was the data controller and she and the first and second authors had access to the fully anonymised dataset.

A report outlining the initial findings of this study has been independently published (i.e., not peer-reviewed) online by Bainne Beatha [33]. This published report predominantly focuses on the quantitative findings but includes a summary of the results in the present paper.

## Population

Individuals over the age of 18, who had a child between January 2019 and December 2021 in an Irish maternity unit or at home in Ireland, and breastfed or considered breastfeeding were eligible to participate. Breastfeeding, in the context of the survey, referred to all types of feeding with breast milk.

## Questionnaire design

The online questionnaire began with a participant information page, at the end of which participants were explicitly requested to provide their consent to participate. After being provided with a detailed participant information page, to proceed to complete the questionnaire, participants had to tick a box stating, "I give informed explicit consent to have my data processed as part of this research study." Questions were created collaboratively by researchers in TU Dublin and Bainne Beatha. The questionnaire consisted of both closed-ended and open-ended questions. No identifying data were collected. The data for the present study was drawn from responses to the following open-ended question: 'if you have any other comments about the breastfeeding support you received in the maternity unit or during your home birth, please add them here.'

## Data collection

The self-administered online questionnaire was made available online using SurveyMonkey[TM] (http://www.surveymonkey.com), a web-based survey platform, from January 24[th], 2022, until February 15[th], 2022, and was shared *via* social media networks (i.e., Twitter and Facebook); thus, our initial strategy was to recruit a convenience sample without aiming to recruit participants with any specific characteristics. However, after 2 weeks of data collection, efforts were made to ensure a national (though not nationally representative) sample was recruited; the hospitals at which women reported giving birth were explored and targeted recruitment of participants from hospitals with the lowest representation was implemented. Thus, our strategy modified into purposeful recruitment. This involved specific social media posts encouraging participants who had delivered at specific hospitals to complete the questionnaire and encouraging politicians local to those hospitals to share these social media posts.

## Data analysis

Data were downloaded from SurveyMonkey[TM] and saved as an SPSS file. The quantitative data (participant characteristics) were analysed in SPSS (version 28) and are presented as frequencies and percentages. A preliminary descriptive analysis of additional quantitative survey data can be found elsewhere [33]. The qualitative data analysis was conducted within an interpretivist paradigm, acknowledging that we can only understand someone's reality through their experience of that reality [34], but no specific theory guided the data analysis. Qualitative data analysis was managed using NVivo Version 12 and followed the 'six-step Thematic Analysis Framework' developed by Braun and Clarke [35]. Despite collecting our data by an online questionnaire, we received many detailed responses to the open-ended question of interest. The average response was 54 words in length and we feel this provided considerable data with which to conduct our thematic analysis.

Following the Framework, data were read analytically by the first author multiple times to become familiar with the content and capture ideas of interest. The content was then coded, at both a semantic and latent level. A revision of the codes was carried out to ensure the coding sets were robust. This revision allowed for the addition to existing, and the creation of new latent codes. Initial themes were developed through compiling the coded data. Developed themes were then reviewed to ensure that they were representative of the dataset and that the research question was being addressed. Themes were defined and named to ensure that they followed a clear direction and were relevant. Finally, written analytical pieces were compiled together and edited to produce the written report. The analysis was carried out in consultation with a second, senior author throughout the process, but all qualitative analysis was completed by one researcher: a female in her early twenties with no children, in her penultimate year of undergraduate study studying a Bachelor of Science degree in Public Health Nutrition. The senior (last) author of the paper is a healthcare professional but does not work in a clinical environment. She, along with several of the other authors, has experience of the maternity care system in Ireland as a service user. As such, the data were collected from this lens.

## Results

### Sample characteristics

There were 5,412 valid responses to the questionnaire and 2,264 of those (41.8%) responded to the open-ended question about their experiences of breastfeeding supports in the maternity unit (**Table 1**). Approximately half were aged 31–35 years, and most were Irish. Just over half gave birth in a maternity unit in Dublin and 53% gave birth in the year 2021. Nearly two thirds (65%) reported having disposable income above their needs at the time of childbirth, with only 1.5% reporting that they were barely making ends meet. Nearly 56% of participants reported that they had no previous breastfeeding experience, and the majority planned to exclusively breastfeed prior to childbirth. Most fed their child directly at the breast for their first feed. Exclusive breastfeeding remained the most common method of feeding 48 hours postpartum (i.e., nothing else consumed since birth), with an increase in the use of formula, both on its own and in combination with breastfeeding.

### Thematic analysis

Two central themes, with associated sub-themes were created from the data: (i) Breastfeeding support in theory but not in practice (ii) Support was either inaccessible, inadequate and/or inappropriate. **Table 2** outlines the structure of the themes and sub-themes generated, along with a theme summary.

### Theme 1: Breastfeeding support in theory but not in practice

This theme describes how breastfeeding was promoted antenatally and how, prior to childbirth, breastfeeding support was portrayed as more available in the maternity unit. As a result, participants expected they would be supported with breastfeeding postpartum and often expressed disappointment that they were not.

A lack of tangible resources was frequently described, in particular, participants emphasised the difficulty in accessing a breast pump or lack of guidance on how to use a pump. Participants reflected on how a lack of tangible resources increased the workload on midwives and praised them for working under such busy conditions, with one participant emphasising how 'midwives do a lot with limited resources' (Participant 5250).

**Table 1. Participant characteristics (n = 2,264).**

| Characteristic | n (%) |
|---|---|
| Age (years) | |
| 18–25 | 17 (0.8) |
| 26–30 | 214 (9.5) |
| 31–35 | 1119 (49.4) |
| 36–40 | 786 (34.7) |
| 41–45 | 121 (5.3) |
| 46 + | 6 (0.3) |
| Country of Origin | |
| Ireland | 2096 (92.7) |
| Other | 163 (7.2) |
| I'd rather not say | 1 (0.0) |
| Where Participant Gave Birth | |
| Dublin | 1134 (54.7) |
| Rest of Leinster | 192 (9.3) |
| Munster | 436 (21.0) |
| Connacht | 200 (9.6) |
| Ulster | 63 (3.0) |
| At home | 44 (2.1) |
| Other | 3 (0.1) |
| I'd rather not say | 2 (0.1) |
| Year of Child's Birth | |
| 2019 | 365 (16.1) |
| 2020 | 696 (30.8) |
| 2021 | 1200 (53.1) |
| Socio-Economic Status | |
| You and your family were barely making ends meet | 35 (1.5) |
| You had some but limited disposable income | 729 (32.2) |
| You had disposable income above your needs | 1471 (65.0) |
| I'd rather not say | 27 (1.2) |
| Previous Breastfeeding Experience | |
| Had previous breastfeeding experience | 906 (40.0) |
| No, this was my first child | 1258 (55.6) |
| No, I have another child but never breastfed | 28 (1.2) |
| Other | 72 (3.2) |
| Antenatal Feeding Plan | |
| Exclusive breastfeeding | 1875 (82.9) |
| Combination feeding (breast milk and formula) | 214 (9.5) |
| Formula feeding | 9 (0.4) |
| I didn't have a plan/unsure | 108 (4.8) |
| Other | 57 (2.5) |
| Baby's First Feed | |
| Child was fed directly at the breast | 1877 (83.1) |
| Child was fed harvested colostrum | 173 (7.7) |
| Child was fed donor breast milk | 4 (0.2) |
| Formula | 112 (5.0) |
| I'm not sure | 14 (0.6) |
| Other | 79 (3.5) |

(*Continued*)

Table 1. (Continued)

| Characteristic | n (%) |
|---|---|
| How Was Baby Fed in the First 48 hours? | |
| Exclusive breastfeeding | 1335 (59.1) |
| Combination feeding | 681 (30.2) |
| Formula | 46 (2.0) |
| Other | 195 (8.6) |

[1]The valid percentages are reported in Table 1 as some data were missing. The variable with the most missing data was 'Where Participant Gave Birth' (n = 190), with all other variables having an average of 3 missing data points.

'I found it confusing as antenatally all the mid wives talked about was breast is best and then once the baby arrived formula wad (sic) pushed despite me insisting I wanted to breast feed, I was post operative and couldn't move in the bed and I watched a midwife give my baby their first feed, I had no support or advocate with me due to the start of the pandemic. . .' (Participant 5850).

'I needed a pump as baby had jaundice so had to triple feed. I don't understand why I had to pay over €100 to rent one when formula is free' (Participant 5588).

Situations where participants were separated from their baby after birth were described, discouraging skin-to-skin contact. Participants noted that bed sharing was not always possible for some as they found it couldn't be done safely/comfortably in the maternity unit.

'Skin to skin was not offered in surgery, there was a rushed feeling due to COVID and it affected my bond with baby' (Participant 1808).

'I only stayed in the hospital 1 night. I wanted to get home to my own bed as quickly as possible as I like to bed share abs (sic) feed baby in bed. This is not possible or comfortable in a hospital bed and hinders the breast feeding experience in my opinion. . . .' (Participant 3607).

Participants who received helpful and need-specific supports used words like 'amazing', 'comforting' and 'motivating' when describing their encounter with the HCP. Many felt that time and staffing constraints acted as a barrier to more one-to-one, positive encounters with HCPs.

Table 2. Thematic analysis structure summary.

| Theme Title | Summary of Theme | Sub-Themes |
|---|---|---|
| Breastfeeding support in theory but not in practice | This theme portrays how breastfeeding was encouraged and promoted antenatally by healthcare professionals. However, the challenges associated with breastfeeding were rarely mentioned antenatally. Women then felt unprepared and unsupported by healthcare professionals in overcoming these challenges postnatally. | 1. Breastfeeding problems were not given breastfeeding solutions<br>2. Emphasis on formula in maternity unit<br>3. Poor knowledge and conflicting advice from healthcare professionals |
| Support was either inaccessible due to lack of staff/time, inadequate; i.e., unhelpful or non-specific and/or physically inappropriate | This theme captures how supports in the maternity unit were mostly unavailable (due to shortages of staff or time), and how when available, they were unhelpful or non-specific, and/or were physically inappropriate. The effect the COVID-19 pandemic had on the availability of supports is included in this theme. | 1. Inaccessible support due to lack of staff/time<br>2. Inadequate; i.e., unhelpful or non-specific support<br>3. Physically inappropriate support |

'I definitely felt supported by the midwives, i.e. they showed me how to express colostrum and gave me a syringe to collect it. However, I really struggled and some of the nights the midwives seemed a lot more stretched.' (Participant 1007).

Sub-theme 1.1: Breastfeeding problems were not given breastfeeding solutions

Many participants described how HCPs did not offer breastfeeding solutions to their breastfeeding problems, which often discouraged them from continuing to breastfeed. Many women mentioned how issues with latch were often left unresolved, leading to extreme pain, and bruising of the nipple. Some described how HCPs said that the pain they were experiencing when breastfeeding was 'normal' and 'would get easier'. Many referred to tongue tie (TT), undiagnosed in the maternity unit, as the underlying cause for issues with latch and the intervention of a private IBCLC was frequently sought to resolve the issue of TT. Many women reported wishing that TT was routinely checked for.

'She wouldn't latch, we tried everything but a lactation specialist was never offered, they offered formula.' (Participant 1821).

'My babies tongue tie was never assessed in the hospital. It was like they didn't believe me that I was in pain. It felt like an electric shock from my nipples to my toes' (Participant 5195).

Participant's nipples/breasts were sometimes referred to as unsuitable for breastfeeding by some HCPs. Some commented on experiencing both low- and oversupply issues with breast milk and feeling like the issues were 'fobbed off' by HCPs.

'Saw the hospital lactation consultant who looked at my feeding attempts from door and told me I would have to pump due to flat nipples. . . .' (Participant 5026).

'I had a massive haemorrhage post delivery and this affected my milk supply.no one spoke about this in the hospital to me. . .' (Participant 3782).

The lack of solutions to breastfeeding problems often left participants feeling 'worried' and 'stressed'. Some ceased breastfeeding due to their problems not being met with appropriate solutions.

'You're so vulnerable in those moments, to be told you're failing, without any solutions offered apart from formula, is just soul crushing. . .' (Participant 4602).

'. . .midwives pressured be (sic) to feed baby formula and told me breastfeeding was very tough and probably not meant for me which led to an emotional breakdown for me in the hospital' (Participant 1105).

Conversely, participants who were assisted in overcoming their breastfeeding problems emphasised how it motivated them to continue breastfeeding.

'I was very lucky with one of the midwives who was on night duty on 3 of the nights I was in. She would wake me up to encourage me to hand express while baby was in NICU and showed me how to do it as I was clueless. It's because of her I kept at it' (Participant 3195).

'I had excellent care with my first baby in 2019 also. She was in neo natal for a week, and they encouraged me to pump, helped me do it, and then helped me feed her. So encouraging.' (Participant 3947).

Sub-theme 1.2: Emphasis on formula in maternity ward

Formula was regularly offered as a solution to the problems described above, with top-ups routinely encouraged by HCPs. Many described the emphasis on formula in the maternity ward using negative language, such as; 'extremely traumatic' and 'pressuring'. The use of formula was often described as 'giving in' and done 'reluctantly'. Some described feeling like they had to leave the hospital to avoid formula feeding.

'I really did not want to give my baby formula but the midwife kept pushing it. She told me my baby was dehydrated because her lips were peeling. This terrified so I caved an gave her formula (I could t (sic) face it I was crying so much with the guilt so the midwife fed her). On reflection I realised the baby's lips were not peeling, it was the lanolin cream from my nipples. I still remember that with such anger. They really pushed formula on me and my baby' (Participant 2929).

'I knew if I didn't get out of the hospital I would end up formula feeding my baby as it seemed everyone had given up' (Participant 3558).

Some participants highlighted how HCPs used their baby's weight gain/loss as an indicator for their health, and how formula top-ups were encouraged to avoid situations of low blood sugar or dehydration due to weight loss. One participant described feeling 'bullied into formula feeding [her] child' due to a weight decrease of 8% 'from birth to day 3' (Subject 2805).

'. . .By my 3rd night in hospital my baby had lost 14% of her body weight and I was told that I had to introduce formula. I was not offered a breastpump; I had to ask for one which led to triple-feeding and ultimately to combination feeding. I never managed to get back to exclusive breastfeeding. . .' (Participant 2491).

Few participants described situations in which breastfeeding was encouraged once issues like low blood sugars, dehydration etc. were resolved using formula.

'Formula was recommend (sic) by midwives as my baby was born premature and had low sugars, once their sugar levels rose, breastfeeding exclusively was fully supported.' (Participant 2641).

Situations where formula was given without the participant's consent or wishes were also described. Women reported feeling vulnerable as a result.

'I was actively discouraged from breastfeeding my twins. They were given formula against my wishes immediately after birth. . .' (Participant 1041).

'The initial midwife had a bottle of formula given to my baby before I knew what was happening. . .' (Participant 1197).

Participants who agreed to formula top-ups often described how they exclusively breastfed once they left the hospital. Others were unable to wean their child off formula and continued combination feeding at home.

'In the hospital she was given formula even though her blood sugars were fine and she was feeding well from the breast. She did not take any formula at home (day 4 onwards) and her

weight gain was perfect for her centile. I now know that there was NO need to give her formula and I really regret that she had it so early on- when she should have been receiving just colostrum' (Participant 4390).

'I wanted to breastfeed but was told I had to give my baby formula due to jaundice. When I questioned this I was told that otherwise baby might have to go under the lights and this was a better option. I feel this really hindered my breastfeeding and I didn't know how to combination feed or cut out the formula once it was decided baby was no longer jaundiced.' (Participant 2538).

Sub-theme 1.3: Poor knowledge and conflicting advice from healthcare professionals

A general lack of knowledge in the maternity unit relating to breastfeeding was commonly described. Hence, many paid for private care from an IBCLC. Advice from HCPs who were particularly interested in breastfeeding, was referred to as more helpful and realistic.

'. . .I found there to be such gaps in knowledge and conflicting answers depending on the age of who I asked. I think staff training is really necessary when it comes to breastfeeding. . .' (Participant 2695).

'Support is very much dependent on the nurse u get and its pot luck. . .nothing standard about their approach' (Participant 1026).

Poor HCP knowledge left many feeling 'confused', 'stressed' and unsure about what advice to follow. Advice on breastfeeding positions, latching and paraphernalia were common areas that participants received conflicting information about.

'Conflicting information from pediatrician (sic), midwifes and lactation consultant about how many wet/dirty nappies were expected, had no confidence in their advice' (Participant 2202).

The emotional stress brought on by conflicting information was expressed by many women, which was felt to be overlooked by HCPs.

'. . .while I received support the support at times was contradictory and changing which was stressful. My stress regarding BF was also not recognised. It felt like my worry was disregarded' (Participant 1332).

'I was given conflicting advice about positions. One midwife told me to lie with my baby and co-sleep while they fed and another midwife find me doing this and chatistised (sic) me and told me I'd smoother my baby. It was very upsetting' (Participant 3843).

Participants who *did* receive encouragement and emotional support from knowledgeable HCPs spoke positively about their experience. Some highlighted how the encouragement they received acknowledged their effort enabled them to continue breastfeeding.

'One midwife was very knowledgable (sic) and encouraged me to breastfeed even when I was feeling low. I worried my nipples were too big for my babies mouth but she assured me my body was perfect for my baby. . .' (Participant 1265).

'. . .I remember one midwife saying 'wow!! Look at all that milk!' it was like 0.2ml but I felt amazing! Its only in hindsight I realised it wasn't a lot but gave me the encouragement I needed.' (Participant 1848).

### Theme 2

**Support was either inaccessible due to lack of staff/time, inadequate; i.e., unhelpful or non-specific and/or physically inappropriate.** Participants highlighted that there was 'no support', and words like 'awful' and 'traumatic' were often used when describing supports. Many felt that breastfeeding was discouraged, particularly if their situation wasn't that of the healthy mother with a healthy baby.

'I requested a lactation consultant 8 times over two days, with two written referrals. No one ever turned up. I was incredibly disappointed and felt completely abandoned' (Participant 4814).

'Very little support especially in first 24 hours resulting in baby only feeding 3 times in first 24hrs, my blood runs cold when I think back on it now' (Participant 2757).

Many participants described how limitations on visiting hours due to the COVID-19 pandemic led to difficulty with breastfeeding and created feelings of loneliness and stress. Participants described how the absence of their partners increased the workload for HCPs as tasks like lifting their baby, typically carried out by a partner/family member, now needed assistance from a HCP.

'Midwives were exceptionally busy as no partners were allowed to visit during covid. I had an emergency section and was told to drink plenty of water. I pressed the call button for water as I could not move my legs and get it myself. I was waiting for 3 hours just to be brought a single cup of water. . .' (Participant 2262).

While it was acknowledged that staff were busy, support was often described as 'non-existent'. Some felt that the only solution was to be discharged/leave the hospital, reporting how family and local GPs would bring more support than the maternity unit.

'I discharged myself signing the against medical advice form so that I could use a pump and not give up breastfeeding. . .' (Participant 5746).

Conversely, participants who received encouragement and practical assistance from HCPs described how it often had a positive influence on their breastfeeding journey, using words like 'encouraging' and 'motivating' to describe the positive supports received.

'The midwives were superb, very encouraging and helpful. Without them I may not have continued to feed' (Participant 1614).

Although only a small number of participants who took part in the survey had a home birth, the majority commented on how positive the experience was. Words like 'wonderful' and 'helpful' were used when describing supports for breastfeeding during their homebirth.

'My homebirth midwife was amazing. This was my 3rd baby- first homebirth and it was the most supportive for breastfeeding overall' (Participant 3029).

Sub-theme 2.1: Inaccessible support due to lack of time/staff
Participants frequently referred to HCPs as 'over-stretched', which made supports 'unavailable' and 'limited'. While some described HCPs as supportive, their lack of time to spend with them left many feeling hopeless.

'Unfortunately the midwives do not have the time to sit with you. . .I felt so frustrated that i just gave up after a few days' (Participant 1306).

'Midwives are well meaning but just don't have time, they are way too understaffed. If you are vulnerable or unsure it would cause you to fall at the first feeding hurdles and give up' (Participant 1360).

Many participants described the difficulty in accessing an IBCLC as they were either too busy or not on site. Specifically, no access to IBCLCs on weekends was highlighted as a major issue.

'LC's [IBCLCs] were too busy, not available when needed, don't work weekends or shifts. Bf'ing [breastfeeding] isn't a mon-fri 9–5 job. It's 24–7 and there should always be a midwife on duty who is a bf specialist' (Participant 1237).

It was often described that when support was needed, it had to be requested because staff were so busy. Situations where vulnerable mothers and those who lack confidence avoided asking for assistance were described. Multiparous participants were thankful for their confidence built from previous experience which enabled them to ask, and in some cases re-ask, for help. While many with previous experience didn't need support, they felt had it been needed, it wouldn't have been available.

'I asked for help and this was my third child so I was able to ask again when it didn't come at first. Many are very vulnerable and are not able to' (Participant 3326).

'I was extremely lucky in that I didn't need support. However, if I had needed it I would have been concerned about how busy the midwives were and their lack of availability to fully support me' (Participant 1252).

Sub-theme 2.2: Inadequate; i.e., unhelpful or non-specific support

When support was given, it was often referred to as impractical and not supporting the participant's specific needs. For example, participants reported that issues with getting the baby to latch were met with no guidance/demonstration. Instead, the midwife/IBCLC latched the baby themselves without any additional information. The need for advice to be more specific was frequently highlighted.

'They were "supportive" in terms of encouraging me. They were not HELPFUL in terms of providing solutions. . .' (Participant 2690).

'It seemed like they wanted to help but didn't have the right, practical knowledge to actually be helpful' (Participant 2373).

'The advice given was general, not specific to me or my baby. . .' (Participant 1970).

Many participants felt consultations with IBCLCs were 'rushed' and that no actual assessment was carried out, and individual's concerns weren't addressed (i.e., assessment of the latch). Many described having to pay for a private IBCLC consultation for specific advice.

'Met with lactation consultant in hospital but she just ran through a document with me; did not assess my baby's tongue tie and did not assess or help with latch' (Participant 3934).

'If I was not adamant and if I didn't have the means to see a private Lactation consultant, I won't be breastfeeding today!. . .' (Participant 5403).

Sub-theme 2.3: Physically inappropriate support

While comments aligned with this sub-theme appeared less frequently, it is salient given the emotionally intense language used throughout. Support was referred to as 'extremely forceful' and 'rough'. Some participants described situations where midwives were 'grabbing' their breast and 'shoving' it in their baby's mouth. One participant described being; 'manhandled by one nurse which left me with bruises on my chest and breasts' (Participant 3983). Similar experiences were described by a small number of mothers:

'. . .I almost gave up after being given out to by a HCA [healthcare assistant] in the public ward and seeing her manhandle my baby like she was a rag doll.' (Participant 4151).

'I'm still traumatised by my experience with midwives pushing my screaming baby's little face into my breast to feed. And then leaving formula on the table as they didn't have any more time to help. I have Ptsd as a result' (Participant 4811).

## Discussion

Breastfeeding supports were generally described as limited and unavailable. When supports were received, participants preferred practical and tailored assistance. Although only a minority of participants referred to supports as physically inappropriate, it emphasises the need for respect and sensitivity in interactions about breastfeeding. A lack of compliance with global breastfeeding standards throughout the maternity units impeded women's ability to establish breastfeeding. Furthermore, insufficient breastfeeding knowledge across all HCPs and limited availability of tangible resources made initiation and continuation of breastfeeding challenging among this sample.

Participants describing breastfeeding supports as limited and unavailable is consistent with previous Irish research which reported a lack of available supports and limited staffing [17]. Participants in the present study recognised how HCPs, particularly midwives, experienced an increased workload due to the implications of COVID-19. Similarly, women in the UK [36, 37] and Belgium [38] reported a lack of face-to-face interactions with HCPs during COVID-19 which impeded their ability to establish breastfeeding. Previous national [39, 40] and international [41] research carried out before COVID-19, which aimed to summarize HCPs' views on BF supports available, identified time [39–41], funding [39] and knowledge [39–41] constraints as significant barriers to aiding women in successfully establishing breastfeeding. The limited availability of supports from HCPs, coupled with the impacts of COVID-19 added significant difficulties to the beginning of women's breastfeeding journeys.

Supports received were described as impractical and non-specific in this present study. Previous research highlights how women value practical demonstrations [42, 43] and information that is adapted to meet their needs [30, 42]. A Cochrane review identified how tailored supports, which meet mothers' specific needs, increase the duration and exclusivity of breastfeeding [30]. The healthcare professionals best placed to meet these needs in Ireland are midwives, lactation consultants, and public health nurses. Unfortunately, maternity units are currently understaffed and results of a recent survey by the Irish Nurses and Midwives Organisation indicate that almost 85% of nurses and midwives said staffing levels could not meet work demands [44]. Staff shortages are recognised by service users as a factor impacting the breastfeeding-related care they receive; over one third of the respondents to this survey indicated

that midwives appeared to be too busy to give them the breastfeeding support they needed [33].

A feasibility study assessing the impact of a multidimensional breastfeeding-support intervention in Ireland included at least one guaranteed consultation with an IBCLC prior to hospital discharge [45]. This consultation was perceived by participants as the most helpful component of the intervention, highlighting the importance of access to in-person support. Consistent with previous research [42, 46], the lack of access to IBCLCs for breastfeeding supports was frequently mentioned in the present study. Intervention from a private IBCLC was described and often occurred post-discharge from hospital, at which point many women had already ceased to exclusively breastfeed. This highlights how supports need to be timely within the initial few days postpartum for the successful initiation and maintenance of breastfeeding [41]. It is pertinent to note that the Irish government allocated funding for the recruitment of 24 additional IBCLCs to the Irish workforce in 2021 to mitigate the scarcity within the public sector [47]. The latest information suggests that to date two IBCLCs have entered the workforce since the implementation of this funding [48]. In-person support across the continuum is viewed as important in Ireland with mothers valuing support groups led by volunteer organisations [32], and those run by public health nurses [49].

While only a small number of participants described receiving rough treatment, it is an alarming finding that highlights the need for supports to be delivered in a sensitive manner. In agreement with the present study, previous studies that examined women's perceptions and experiences of breastfeeding supports reported that, while a 'hands-on' approach by HCPs is beneficial, it is sometimes viewed as intrusive [34], particularly if it was perceived as 'heavy-handed' [50]. Schmied and colleagues also reported how women felt their breasts were seen as a 'feeding implement' by HCPs [34]. Failure to validate women's ability to breastfeed has led to feelings of inadequacy as a woman [50]. It is therefore further distressing that women in the present study were led to believe that their nipples/breasts were unsuitable for breastfeeding by HCPs. These findings, along with those describing infant formula being provided against a mothers' wishes, indicate that in some cases women are not being treated with dignity and respect. These experiences highlight the need for the re-establishment of an environment that is supportive, compassionate and respectful towards those who wish to breastfeed.

Contrary to international and national recommendations, participants in this study described how they were separated from their baby post-birth, limiting the act of skin-to-skin contact. Skin-to-skin aids in the development of the mother-infant dyad through encouraging attachment, which has shown to increase rates of breastfeeding [51]. In addition, participants in the present study described how infant formula was readily available and often used as a solution to breastfeeding challenges. This is mirrored in the NMES, where women described how midwives offered formula top-ups rather than assistance [17]. Consistent with previous research [52], the use of terms such as 'giving in' and feelings of guilt were associated with the use of formula. The recent re-introduction of the BFHI [26] and the policy on Marketing of Breast Milk Substitutes [27] may lead to positive changes regarding formula availability and use throughout maternity units. With women spending on average 2 days post-birth in Irish maternity units [53], full implementation of the Code within maternity units is essential to create a positive environment to support breastfeeding.

The lack of tangible resources, such as breast pumps and other paraphernalia, made it difficult to deal with challenges relating to breastfeeding such as pain and perceived lack of milk supply. Some of these issues are recognized as causing early cessation of breastfeeding [54, 55], highlighting how important it is to have these items available. Again, these findings emphasise the need for individualised, timely support that offers breastfeeding solutions to breastfeeding problems.

This research has clinical and policy implications. HCPs need to have adequate knowledge through sufficient training to assist women in overcoming breastfeeding challenges, which has been previously suggested [40], and they need to have time to provide tailored advice and support. Recent evidence suggests that some allied HCPs in Ireland would welcome such training [56]. The findings support the need for increased assistance from IBCLCs. Therefore, a more positive perception of supports may be expected with the number of IBCLCs anticipated to join the Irish workforce in the near future [47]. Additionally, incorporation of the Ten Steps to Successful Breastfeeding could allow for increased rates of breastfeeding initiation and exclusivity [57].

## Strengths and limitations

The use of open-ended questions allowed participants to express their feelings and provide information that would not have been obtained with closed-ended questions with investigator-generated responses. Preliminary data were presented at a conference of the Association of Lactation Consultants of Ireland and highlights were posted on social media. Feedback received suggests that our findings are reflective of the experiences of those who receive breastfeeding-related care in Ireland.

However, using an online questionnaire to collect qualitative data can limit the richness of the responses obtained; an in-depth interview or request for a detailed narrative could have provided a better reflection of individual experiences. Additional limitations include the use of a self-reported questionnaire advertised *via* social media was used to collect the data, making selection bias possible as those who don't have access to the internet or social media couldn't participate. Selection bias is likely when noting the demographic characteristics of the cohort; this sample is well educated and of higher socioeconomic status than the general population. The study aimed at recruiting people who breastfed or considered breastfeeding. As expected, a higher proportion of people breastfed in the present study sample compared to the general Irish population. Therefore, the sample isn't representative of the general Irish population. In addition, data were collected using a questionnaire developed by those with experience as service users and were analysed largely from that lens. To obtain a more holistic understanding of the provision of breastfeeding support in Ireland, research from the perspective of healthcare professionals is needed.

A possibility for recall bias must be considered as women relied on memories of their experiences. Data were collected during COVID-19. As a result, the implications of COVID-19 throughout the maternity wards may influence the findings and must be considered when interpreting results. However, the themes generated were considered with the potential impact of COVID-19 in mind. Despite half of the births occurring during 2021, when supports available were impacted by COVID-19, there was no obvious difference in the overall tone of comments when compared to those who experienced support prior to the occurrence of COVID-19.

## Conclusions

Many of our participants did not experience the support required to successfully initiate or continue breastfeeding. Findings of this study emphasise the lack of timely, needs-specific and practical supports. Supports within the postpartum period are essential, and need to be delivered in an individualised, informative and unbiased way in order to build women's confidence to establish exclusive breastfeeding. Although participants appeared receptive to supports for breastfeeding when received, external barriers such as lack of time and staff in the maternity unit meant accessing supports was difficult. Additional staffing and implementation of policy surrounding formula use in the maternity unit would likely lead to improved experiences for breastfeeding parents.

## Acknowledgments

We are immensely grateful to all the participants who took the time to share their experiences with us.

## Author Contributions

**Conceptualization:** Lucia Prihodova, Deborah Byrne, Megan Etherton, Felicienne Rahill, Catie Wilson, Elizabeth J. O'Sullivan.

**Data curation:** Lucia Prihodova, Deborah Byrne, Megan Etherton, Felicienne Rahill, Catie Wilson, Elizabeth J. O'Sullivan.

**Formal analysis:** Niamh Lawlor, Elizabeth J. O'Sullivan.

**Methodology:** Lucia Prihodova, Deborah Byrne, Megan Etherton, Felicienne Rahill, Catie Wilson, Elizabeth J. O'Sullivan.

**Project administration:** Lucia Prihodova, Deborah Byrne, Megan Etherton, Felicienne Rahill, Catie Wilson, Elizabeth J. O'Sullivan.

**Supervision:** Elizabeth J. O'Sullivan.

**Writing – original draft:** Niamh Lawlor.

**Writing – review & editing:** Lucia Prihodova, Deborah Byrne, Megan Etherton, Felicienne Rahill, Catie Wilson, Elizabeth J. O'Sullivan.

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
