## [Decision Letter · Decision Letter 0]

6 Apr 2023

PONE-D-23-04748A Qualitative Analysis of Women’s Postnatal Experiences of Breastfeeding Supports During the Perinatal Period in IrelandPLOS ONE

Dear Dr. O'Sullivan,

Thank you for submitting your manuscript to PLOS ONE. After careful consideration, we feel that it has merit but does not fully meet PLOS ONE’s publication criteria as it currently stands. Therefore, we invite you to submit a revised version of the manuscript that addresses the points raised during the review process.

We look forward to receiving your revised manuscript.

Kind regards,

Kyaw Lwin Show, MPH

Academic Editor

PLOS ONE

Journal Requirements:

3. We noted in your submission details that a portion of your manuscript may have been presented or published elsewhere. [A report outlining the initial findings of this study has been published online by Bainne Beatha (Bainne Beatha and O’Sullivan, 2022). This report predominantly focuses on the quantitative findings, but includes a summary of the results in the present paper.] Please clarify whether this [conference proceeding or publication] was peer-reviewed and formally published. If this work was previously peer-reviewed and published, in the cover letter please provide the reason that this work does not constitute dual publication and should be included in the current manuscript.

Reviewers' comments:

Reviewer's Responses to Questions

**Comments to the Author**

1. Is the manuscript technically sound, and do the data support the conclusions?

Reviewer #1: Yes

Reviewer #2: Yes

2. Has the statistical analysis been performed appropriately and rigorously? 

Reviewer #1: N/A

Reviewer #2: N/A

3. Have the authors made all data underlying the findings in their manuscript fully available?

Reviewer #1: No

Reviewer #2: No

4. Is the manuscript presented in an intelligible fashion and written in standard English?

Reviewer #1: Yes

Reviewer #2: Yes

5. Review Comments to the Author

Reviewer #1: Manuscript Number: PONE-D-23-04748

Thank you for the opportunity to review this article, I appreciate the time and effort the research team has placed in the conduction of this study, writing of this manuscript, and preparing it for submission. This study provides insight on the experiences of breastfeeding supports during the perinatal period in Ireland, it can potentially inform future interventions and programs for the improvement of breastfeeding support in maternity care. Below are specific recommendations for the manuscript text:

Introduction:

Since the study period was conducted before and during the covid19 pandemic (and the impact is mentioned in the findings and discussion) it would be useful to have information on this section on how the pandemic disrupted maternity care services.

Please discuss in the introduction recent relevant evidence on the overall experience of breastfeeding in Ireland, it would be then useful to then compare this study findings to said evidence in the discussion section, examples of recent evidence:

Murphy et al, 2022 - Women's experiences of initiating feeding shortly after birth in Ireland: A secondary analysis of quantitative and qualitative data from the National Maternity Experience Survey. Midwifery,Volume 107,2022,103263,ISSN 0266-6138,https://doi.org/10.1016/j.midw.2022.103263.

Quinn et al, 2019-A qualitative exploration of breastfeeding support groups in Ireland from the women's perspectives ,Midwifery,Volume 78,2019,Pages 71-77,ISSN 0266-6138,https://doi.org/10.1016/j.midw.2019.08.001.

Methods:

Please provide a section on ethical considerations beyond the informed consent, how was confidentiality and privacy addressed, how was data managed? Who had access to the data?

Data collection

Line 126: please reference survey monkey and provide a description of what it is such as “web-based survey platform.”

Line 128: it is not clear why the authors considered the sample to be a convenience sample instead of a purposeful one just because recruitment was done online, if applicable please change to purposeful sample, if not, please clarify.

Data analysis

Line 137 and 138: since quantitative findings are not included in this study, how quant analysis was conducted is irrelevant, it would be more appropriate to mention that quantitative findings have been disseminated elsewhere and include a reference.

It is best practice in qualitative research to have a social theory (e.g, critical race theory, feminism, etc) guiding the analysis and interpretation of findings, was any social theory considered? If so, please specify it in this section.

Line:140-141: reference needed for interpretivist paradigm

Line 151-156: thank you for the positionality statement, do authors have any reflections on any assumptions or structures on power that might have influenced the analysis? If so, how were they addressed? For the sake of reflexivity and best qualitative research practice it would be useful to add a sentence expanding on the above if applicable.

Results

Sample characteristics

There is a repetitive use of the words over half in this whole section, it is ok to use the percentages, please report accordingly.

Thematic analysis

Very interesting and important findings in this section!

Please consider changing the order of theme 1 and theme 2 to improve the logical narrative of the findings, it could be clearer to discuss “breastfeeding support in theory but not in practice” first, and then discuss how that support was not given because “ support was either inaccessible, inadequate and/or inappropriate”.

I suggest selecting the most relevant, representative or impactful quotes and removing others to avoid repetition, and then using the remaining wordcount to provide more commentary on the analysis of the selected quotes.

In the quotes throughout this section please considering using the word participant or a pseudonym instead of subject to avoid dehumanization of women.

For clarity, I suggest re-numbering the subthemes in theme 1 as 1.1, 1.2 and 1.3 and the subthemes in theme 2 as 2.1, 2.2 and 2.3.

Line 246: please edit this quote to clarify the participant is referring to breastfeeding when saying bf’ing and bf as it might not be obvious to all readers

In sub theme 3 inappropriate support, sub theme 1 Breastfeeding problems were given breastfeeding solutions and sub theme 2 emphasis and formula in maternity wards, there seems to be an ongoing theme of mistreatment of women, such as violent disregard for body autonomy of women ( grabbing their breasts, being manhandled), their emotional well-being (referring to their nipples as “ unsuitable”) and personal feeding choices (“giving formula without the participants consent”). This is an important finding from a health equity and women’s reproductive rights perspective, I suggest regrouping the relevant quotes under inappropriate support sub-theme, and or a another subtheme , as it can be argued that mistreatment of women is not really support.

Discussion:

Great discussion! Please take into consideration suggestions about this section on the introduction feedback regarding other relevant evidence in the Irish context.

Line 594-596: Please provide reference that support the claim that the ten steps to successful breastfeeding could allowed for more positive breastfeeding experiences.

Strengths and limitations:

Line 608: as a note, qualitative findings are not meant to be generalizable therefore a representative sample of the population is not needed.

Conclusions:

Line 619: because qualitative findings are not generalizable and the sample was not representative, the assertion in line 619 and 620 cannot be made from the findings of this study alone (and that’s ok! The findings are valid and important), please edit this sentence to explicitly center the experiences of women in this study and not of the country as a whole.

Reviewer #2: Thank you for the opportunity to review this paper which reports qualitative responses to a survey question regarding support during breastfeeding experiences. The survey received many responses which speaks to the demand and interest regarding breastfeeding in the Irish context. The authors have broadly navigated well the challenges of analysing qualitative data collected through a principally quantitative lens (survey). The following feedback is offered to further strengthen this paper.

Study Context/Setting

The introduction provides sufficient information about breastfeeding in the Irish context and the role of the BFHI, thank you. For an international audience, more context is needed on maternity care in Ireland as it is relevant to this study. If the authors could please address the following points:

1. Explain the scope of the midwife in Ireland with respect to breastfeeding and breastfeeding education

a. What training do midwives receive on breastfeeding and supporting women to breastfeed, as this is part of their scope of practice?

b. What involvement do midwives have in the provision antenatal breastfeeding education?

2. Explain routine postnatal care in Ireland

a. How long do women typically stay in hospital postpartum?

b. Is there a postnatal visiting midwifery service that provides further support in the postnatal period following discharge?

For readers unfamiliar with the topic, it would also be of value to compare breastfeeding rates in Ireland with those in similar countries.

Study Design

The stated aim of this study is to describe women’s personal experiences of supports for breastfeeding within the Irish healthcare system during their time in the maternity unit, or during their home birth in Ireland to provide evidence for how support can be improved. To meet this aim, a single open-ended question at the end of a cross-sectional survey was used, “if you have any other comments about the breastfeeding support you received in the maternity unit or during your homebirth, please add them here”. Both the wording of this question and the choice of a survey tool to collect the data limit the richness and depth of experiences that may be captured and the ability of the research team to meet this aim. Please reflect on this as a limitation of the study design.

Similarly, we would encourage the authors to consider if the choice of thematic analysis is appropriate considering the depth of responses to the questions. The inclusion of a few sentences or a paragraph to justify why this approach was chosen is recommended.

We note that the thematic analysis was conducted by one author, overseen by a senior researcher. This is not standard practice in thematic analysis. Please address this as a limitation and consider how it may influence the study findings.

The research findings would be strengthened by explanation of how the confirmability of the research findings was addressed. This may clarify how the findings reflect the participants’ experiences as opposed to the researchers’ potential biases. A paragraph articulating how reflexivity has been considered, including if any of the research team are midwives or women with lived experience, or presenting the findings to women or key stakeholders for feedback/member checking may be of value here.

The description of themes provides a moving snapshot of women’s experiences of breastfeeding supports in Ireland, well done. Please place a participant ID number after all quotes and please do not repeat the same quote multiple times (subject 1041). We would also encourage the authors to reconsider the term, “subject”, and to review the quotes to ensure that they support the description of the themes. Please also review the description of themes and subthemes and consider more clearly delineating them from each other. For example, the subthemes, “Inaccessible support”, “Inadequate support” and “Inappropriate support” are repetitive and not clearly differentiated from one another.

The authors have acknowledged that they moved from a convenience sampling approach to purposive sampling to capture more nationally representative data, as a limitation of this study.

Discussion and Conclusion

Given the low breastfeeding rates in Ireland, breastfeeding support is clearly an important area of research. The findings articulated in this paper show that further research is urgently needed on how healthcare providers can be better supported to, in turn, support women to breastfeed. This is an important contribution to the literature. However, there is a strong emphasis in the framing of this study (introduction and discussion) on the role of IBCLCs. While these IBCLCs are clearly important stakeholders and valued by the women, the provision of breastfeeding support is well within the scope of midwives as indicated in International Confederation of Midwives Essential Competencies. We encourage the authors to consider how midwifery staffing shortages, training needs and systems issues may have contributed to the lack of breastfeeding support reported by the women in this study in their discussion. The authors are also encouraged to consider how these findings may inform future research to address factors influencing poor breastfeeding support.

6. PLOS authors have the option to publish the peer review history of their article (what does this mean?). If published, this will include your full peer review and any attached files.

Reviewer #1: No

Reviewer #2: No

---

## [Author Response · Author response to Decision Letter 0]

22 May 2023

Reviewer #1: Manuscript Number: PONE-D-23-04748

Thank you for the opportunity to review this article, I appreciate the time and effort the research team has placed in the conduction of this study, writing of this manuscript, and preparing it for submission. This study provides insight on the experiences of breastfeeding supports during the perinatal period in Ireland, it can potentially inform future interventions and programs for the improvement of breastfeeding support in maternity care. Below are specific recommendations for the manuscript text:

Introduction:

Since the study period was conducted before and during the covid19 pandemic (and the impact is mentioned in the findings and discussion) it would be useful to have information on this section on how the pandemic disrupted maternity care services.

Thank you for this suggestion. Additional detail has been added to the introduction regarding the impact of Covid-19 on maternity services in Ireland. Please see lines 72-85. 

Please discuss in the introduction recent relevant evidence on the overall experience of breastfeeding in Ireland, it would be then useful to then compare this study findings to said evidence in the discussion section, examples of recent evidence:

Murphy et al, 2022 - Women's experiences of initiating feeding shortly after birth in Ireland: A secondary analysis of quantitative and qualitative data from the National Maternity Experience Survey. Midwifery, Volume 107,2022,103263,ISSN 0266-6138,https://doi.org/10.1016/j.midw.2022.103263.

Quinn et al, 2019-A qualitative exploration of breastfeeding support groups in Ireland from the women's perspectives ,Midwifery,Volume 78,2019,Pages 71-77,ISSN 0266-6138,https://doi.org/10.1016/j.midw.2019.08.001.

Thank you for this suggestion. Murphy et al., had been referenced in the introduction and discussion of the original draft. We have now added a reference to the work of Quinn et al., also (see lines 121-126 and 586-588. 

Methods:

Please provide a section on ethical considerations beyond the informed consent, how was confidentiality and privacy addressed, how was data managed? Who had access to the data?

Thank you for this comment. Additional detail has been added in the 1st paragraph of the methods section. See lines 141-143. 

Data collection

Line 126: please reference survey monkey and provide a description of what it is such as “web-based survey platform.”

This information has been added. See line 170.

Line 128: it is not clear why the authors considered the sample to be a convenience sample instead of a purposeful one just because recruitment was done online, if applicable please change to purposeful sample, if not, please clarify.

Thank you for this request for clarification. Initially, we considered this to be a convenience sample as we cast the net out wide and did not aim to recruit participants with any specific characteristics. However, as time went on, we modified our recruitment strategy to ensure we recruited a sample of women who had delivered at each maternity hospital in Ireland, turning our recruitment strategy into a purposeful one. We have modified the text for clarity. Please see lines 171-177.

Data analysis

Line 137 and 138: since quantitative findings are not included in this study, how quant analysis was conducted is irrelevant, it would be more appropriate to mention that quantitative findings have been disseminated elsewhere and include a reference.

The quantitative analysis referred to here is simply the descriptive characteristics of the sample who contributed data to the qualitative analysis, which are presented in Table 1. We have now included a sentence to signpost readers to the additional quantitative analysis that is available online, thank you. See lines 184-185.

It is best practice in qualitative research to have a social theory (e.g., critical race theory, feminism, etc) guiding the analysis and interpretation of findings, was any social theory considered? If so, please specify it in this section.

Thank you for this query. We did not approach data collection or data analysis with a specific theory in mind. Our goal was to collect a diverse range of experiences and did not use a theory to guide analysis as we did not want to impose a structure on the results. We have highlighted in the first paragraph about data analysis that not specific theory was used to guide the analysis. See line 187.

Line:140-141: reference needed for interpretivist paradigm

Thank you for this query. A reference has been added. See line 187.

Line 151-156: thank you for the positionality statement, do authors have any reflections on any assumptions or structures on power that might have influenced the analysis? If so, how were they addressed? For the sake of reflexivity and best qualitative research practice it would be useful to add a sentence expanding on the above if applicable.

Thank you for encouraging us to delve deeper and practice more reflexively. We have included additional detail about the make-up of the team, highlighting that most have experience of the Irish maternity care system from the perspective of service user. The potential influence of that on data collection and analysis has been acknowledged in the limitations and it is acknowledged that research is needed from the perspective of healthcare professionals so that a holistic study of the problem of poor breastfeeding support in Ireland can be done. See lines 206-209 and 654-657.

Results

Sample characteristics

There is a repetitive use of the words over half in this whole section, it is ok to use the percentages, please report accordingly.

Thank you for this suggestion. This paragraph has been modified to reduce the repetition. See lines 213-218.

Thematic analysis

Very interesting and important findings in this section!

Please consider changing the order of theme 1 and theme 2 to improve the logical narrative of the findings, it could be clearer to discuss “breastfeeding support in theory but not in practice” first, and then discuss how that support was not given because “ support was either inaccessible, inadequate and/or inappropriate”.

Thank you for this suggestion; we agree with your suggestion, it is very helpful to have another perspective to highlight these things. We have changed the order of the themes presented in the results section. We have also expanded upon the names of the subthemes in what is now theme 2, at the request of the other reviewer. We feel that the themes still reflect the data as a whole, but hope they are now presented more clearly. 

I suggest selecting the most relevant, representative or impactful quotes and removing others to avoid repetition, and then using the remaining wordcount to provide more commentary on the analysis of the selected quotes.

Thank you for this suggestion, we have removed a small number of quotations and feel that the remaining quotations are a good reflection of the themes described. 

In the quotes throughout this section please considering using the word participant or a pseudonym instead of subject to avoid dehumanization of women. 

Thank you for this suggestion, this change has been made throughout. 

For clarity, I suggest re-numbering the subthemes in theme 1 as 1.1, 1.2 and 1.3 and the subthemes in theme 2 as 2.1, 2.2 and 2.3. 

Thank you for this suggestion, this change has been made.

Line 246: please edit this quote to clarify the participant is referring to breastfeeding when saying bf’ing and bf as it might not be obvious to all readers. 

Thank you for this request for clarity; this change has been made. See line 484. 

In sub theme 3 inappropriate support, sub theme 1 Breastfeeding problems were given breastfeeding solutions and sub theme 2 emphasis and formula in maternity wards, there seems to be an ongoing theme of mistreatment of women, such as violent disregard for body autonomy of women ( grabbing their breasts, being manhandled), their emotional well-being (referring to their nipples as “ unsuitable”) and personal feeding choices (“giving formula without the participants consent”). This is an important finding from a health equity and women’s reproductive rights perspective, I suggest regrouping the relevant quotes under inappropriate support sub-theme, and or a another subtheme, as it can be argued that mistreatment of women is not really support. Thank you for this suggestion. Having re-worked the results slightly (I.e., changed the order of the themes and modified the names of the subthemes in theme 2) we feel that the current thematic structure is a good representation of the data. However, we agree with your point regarding the mistreatment of women and have incorporated these points into the discussion. Please see lines 594-603 in the discussion. 

Discussion:

Great discussion! Please take into consideration suggestions about this section on the introduction feedback regarding other relevant evidence in the Irish context.

Ref Murphy and Quinn...

Thank you very much for the positive feedback and for this suggestion, more detail has been added in the discussion, referencing also Leahy-Warren and colleagues 2017 (doi: 10.1016/j.wombi.2016.10.002). See lines 586-588.

Line 594-596: Please provide reference that support the claim that the ten steps to successful breastfeeding could allowed for more positive breastfeeding experiences.

Thank you for this recommendation. We have modified this sentence slightly to indicated that incorporating the 10 steps could increase breastfeeding initiation and duration rates and have included a reference for same. Please see lines 632-634.

Strengths and limitations:

Line 608: as a note, qualitative findings are not meant to be generalizable therefore a representative sample of the population is not needed.

Thank you for this comment. We think it may be likely that others may think the study is representative of the whole population, given our large sample size so perhaps it is still worth noting. 

Conclusions:

Line 619: because qualitative findings are not generalizable and the sample was not representative, the assertion in line 619 and 620 cannot be made from the findings of this study alone (and that’s ok! The findings are valid and important), please edit this sentence to explicitly center the experiences of women in this study and not of the country as a whole. 

Thank you for highlighting this. The first sentence of the conclusions has been modified to reflect the experiences of the women in our study. Please see lines 669-670.

Reviewer #2: Thank you for the opportunity to review this paper which reports qualitative responses to a survey question regarding support during breastfeeding experiences. The survey received many responses which speaks to the demand and interest regarding breastfeeding in the Irish context. The authors have broadly navigated well the challenges of analysing qualitative data collected through a principally quantitative lens (survey). The following feedback is offered to further strengthen this paper.

Study Context/Setting

The introduction provides sufficient information about breastfeeding in the Irish context and the role of the BFHI, thank you. 

For an international audience, more context is needed on maternity care in Ireland as it is relevant to this study. If the authors could please address the following points:

1. Explain the scope of the midwife in Ireland with respect to breastfeeding and breastfeeding education

a. What training do midwives receive on breastfeeding and supporting women to breastfeed, as this is part of their scope of practice? 

b. What involvement do midwives have in the provision antenatal breastfeeding education?

2. Explain routine postnatal care in Ireland

a. How long do women typically stay in hospital postpartum?

b. Is there a postnatal visiting midwifery service that provides further support in the postnatal period following discharge? 

For readers unfamiliar with the topic, it would also be of value to compare breastfeeding rates in Ireland with those in similar countries.

Thank you for this request for clarity for an international audience. We have added additional detail (in red) at the end of the first paragraph of the introduction to compare breastfeeding in Ireland with other European countries and the US; see lines 43-45. 

We have also added information about standard midwifery and postnatal care; please see lines 50-52 and 55-60.

Study Design

The stated aim of this study is to describe women’s personal experiences of supports for breastfeeding within the Irish healthcare system during their time in the maternity unit, or during their home birth in Ireland to provide evidence for how support can be improved. To meet this aim, a single open-ended question at the end of a cross-sectional survey was used, “if you have any other comments about the breastfeeding support you received in the maternity unit or during your homebirth, please add them here”. Both the wording of this question and the choice of a survey tool to collect the data limit the richness and depth of experiences that may be captured and the ability of the research team to meet this aim. Please reflect on this as a limitation of the study design.

Thank you for this suggestion, this limitation has been acknowledged. Please see lines 644-646. 

Similarly, we would encourage the authors to consider if the choice of thematic analysis is appropriate considering the depth of responses to the questions. The inclusion of a few sentences or a paragraph to justify why this approach was chosen is recommended.

Thank you for this comment. As now noted in the Data Analysis section, despite collecting our data by an online questionnaire, we received many detailed responses to the open-ended question of interest. The average response was 54 words in length (range 1 to 884). We feel this provided considerable data with which to conduct our thematic analysis. Please see lines 189-192.

We note that the thematic analysis was conducted by one author, overseen by a senior researcher. This is not standard practice in thematic analysis. Please address this as a limitation and consider how it may influence the study findings.

Thank you for this query. In our analysis, we were guided by Braun and Clarke’s 2021 book titled “Thematic analysis: a practical guide.” In this book, they discourage a focus on multiple coders (and aiming for coder reliability) and acknowledge that qualitative research is a subjective process. Instead of focusing on coder reliability, they encourage reflexive thinking and a description of how the analyst’s attributes and values may have influenced the analysis completed. We have expanded on this information in the data analysis section and hope that this detail is sufficient. Please see lines 206-209 and 654-657.

The research findings would be strengthened by explanation of how the confirmability of the research findings was addressed. This may clarify how the findings reflect the participants’ experiences as opposed to the researchers’ potential biases. A paragraph articulating how reflexivity has been considered, including if any of the research team are midwives or women with lived experience, or presenting the findings to women or key stakeholders for feedback/member checking may be of value here.

Thank you for this comment. We have added some additional detail to the data analysis section to highlight that many of the research team have experience of the maternity care system in Ireland as service users. We have acknowledged this as a limitation in the limitations section, highlighting that research could be conducted among healthcare professionals to provide a more holistic view of the topic. We have also included a point about confirmability in the Strengths and Limitations section. These data have been presented at a conference, and informally shared on social media. All feedback so far indicates that our findings are reflective of the true experiences of mothers who obtain breastfeeding-related support in Ireland. Please see lines 206-209, 639-642 and 654-657.

The description of themes provides a moving snapshot of women’s experiences of breastfeeding supports in Ireland, well done. Please place a participant ID number after all quotes and please do not repeat the same quote multiple times (subject 1041). 

Thank you for this positive feedback, and your request for clarification. A participant ID is provided after each of the long, indented quotations. Elsewhere, we have provided short (<5 words) in-text quotations within individual sentences. We feel that adding participant IDs here will make these sentences difficult to read and reduce their impact. As such, we have not made this change. However, if this is necessary, we can either add the participant IDs or remove the quotations and explain the concepts in our own words.

We would also encourage the authors to reconsider the term, “subject”, and to review the quotes to ensure that they support the description of the themes. 

Thank you for this suggestion, the term “Subject” has been modified to “Participant” throughout to avoid the de-humanising of our respondents. Additionally, quotations have been reviewed. We have removed a small number of quotations and feel that the remaining quotations are a good reflection of the themes described. 

Please also review the description of themes and subthemes and consider more clearly delineating them from each other. For example, the subthemes, “Inaccessible support”, “Inadequate support” and “Inappropriate support” are repetitive and not clearly differentiated from one another.

Based on the feedback from the other reviewer, we have changed the order of the themes presented in the results section. We have also expanded upon the names of the subthemes in what is now theme 2 to clearly distinguish them from one another. We feel that the themes still reflect the data as a whole, but hope they are now presented more clearly. Changes are highlighted in red in the text. 

The authors have acknowledged that they moved from a convenience sampling approach to purposive sampling to capture more nationally representative data, as a limitation of this study.

Additional detail about our sampling approach has been added. Please see lines 171-177).

Discussion and Conclusion

Given the low breastfeeding rates in Ireland, breastfeeding support is clearly an important area of research. The findings articulated in this paper show that further research is urgently needed on how healthcare providers can be better supported to, in turn, support women to breastfeed. This is an important contribution to the literature.

Thank you for this positive feedback.

However, there is a strong emphasis in the framing of this study (introduction and discussion) on the role of IBCLCs. While these IBCLCs are clearly important stakeholders and valued by the women, the provision of breastfeeding support is well within the scope of midwives as indicated in International Confederation of Midwives Essential Competencies. We encourage the authors to consider how midwifery staffing shortages, training needs and systems issues may have contributed to the lack of breastfeeding support reported by the women in this study in their discussion. The authors are also encouraged to consider how these findings may inform future research to address factors influencing poor breastfeeding support.

Thank you for this feedback, we have unintentionally overly focused on one healthcare provider. We have included more detail about the role of midwives in the provision of breastfeeding-related care, and the challenges associated with staff shortages. The need for research from the perspective of the healthcare professional has also been highlighted in the limitations section. Please see lines 55-60, 66-70, 121-124, 565-572, 586-588.

---

## [Decision Letter · Decision Letter 1]

22 Jun 2023

A Qualitative Analysis of Women’s Postnatal Experiences of Breastfeeding Supports During the Perinatal Period in Ireland

PONE-D-23-04748R1

Dear Dr. O'Sullivan,

We’re pleased to inform you that your manuscript has been judged scientifically suitable for publication and will be formally accepted for publication once it meets all outstanding technical requirements.

Kind regards,

Kyaw Lwin Show, MPH

Academic Editor

PLOS ONE

Additional Editor Comments (optional):

Reviewers' comments:

Reviewer's Responses to Questions

**Comments to the Author**

1. If the authors have adequately addressed your comments raised in a previous round of review and you feel that this manuscript is now acceptable for publication, you may indicate that here to bypass the “Comments to the Author” section, enter your conflict of interest statement in the “Confidential to Editor” section, and submit your "Accept" recommendation.

Reviewer #1: All comments have been addressed

2. Is the manuscript technically sound, and do the data support the conclusions?

Reviewer #1: Yes

3. Has the statistical analysis been performed appropriately and rigorously? 

Reviewer #1: N/A

4. Have the authors made all data underlying the findings in their manuscript fully available?

Reviewer #1: No

5. Is the manuscript presented in an intelligible fashion and written in standard English?

Reviewer #1: Yes

6. Review Comments to the Author

Reviewer #1: (No Response)

7. PLOS authors have the option to publish the peer review history of their article (what does this mean?). If published, this will include your full peer review and any attached files.

Reviewer #1: No

---

## [Editor Report · Acceptance letter]

30 Jun 2023

PONE-D-23-04748R1 

A Qualitative Analysis of Women’s Postnatal Experiences of Breastfeeding Supports During the Perinatal Period in Ireland 

Dear Dr. O'Sullivan:

I'm pleased to inform you that your manuscript has been deemed suitable for publication in PLOS ONE. Congratulations! Your manuscript is now with our production department. 

Kind regards, 

on behalf of

Dr. Kyaw Lwin Show 

Academic Editor

PLOS ONE